# NK Cell-Based Immunotherapy in Colorectal Cancer

**DOI:** 10.3390/vaccines10071033

**Published:** 2022-06-28

**Authors:** Mariella Della Chiesa, Chiara Setti, Chiara Giordano, Valentina Obino, Marco Greppi, Silvia Pesce, Emanuela Marcenaro, Mariangela Rutigliani, Nicoletta Provinciali, Laura Paleari, Andrea DeCensi, Simona Sivori, Simona Carlomagno

**Affiliations:** 1Department of Experimental Medicine (DIMES), University of Genoa, 16132 Genoa, Italy; settichiara.90@gmail.com (C.S.); chiara_giordano@live.it (C.G.); valentinaobino@gmail.com (V.O.); marcogreppi92@gmail.com (M.G.); silvia.pesce@unige.it (S.P.); emanuela.marcenaro@unige.it (E.M.); simona.sivori@unige.it (S.S.); 2Pathology Unit, Galliera Hospital, 16128 Genoa, Italy; mariangela.rutigliani@galliera.it; 3Medical Oncology, Galliera Hospital, 16128 Genoa, Italy; nicoletta.provinciali@galliera.it (N.P.); andrea.decensi@galliera.it (A.D.); 4A.Li.Sa., Liguria Region Health Authority, 16121 Genoa, Italy; laura.paleari@regione.liguria.it

**Keywords:** NK cells, monoclonal antibodies, bispecific antibodies, trispecific engagers, immune checkpoints, CAR-NK cells, CRC, immunotherapies

## Abstract

Human Natural Killer (NK) cells are all round players in immunity thanks to their powerful and immediate response against transformed cells and the ability to modulate the subsequent adaptive immune response. The potential of immunotherapies based on NK cell involvement has been initially revealed in the hematological setting but has inspired the design of different immune tools to also be applied against solid tumors, including colorectal cancer (CRC). Indeed, despite cancer prevention screening plans, surgery, and chemotherapy strategies, CRC is one of the most widespread cancers and with the highest mortality rate. Therefore, further efficient and complementary immune-based therapies are in urgent need. In this review, we gathered the most recent advances in NK cell-based immunotherapies aimed at fighting CRC, in particular, the use of monoclonal antibodies targeting tumor-associated antigens (TAAs), immune checkpoint blockade, and adoptive NK cell therapy, including NK cells modified with chimeric antigen receptor (CAR-NK).

## 1. Introduction

Colorectal cancer (CRC) is the third-most-common cancer worldwide and claims almost 1 million deaths per year, despite effective cancer prevention screening plans and relatively good prognosis compared to other gastrointestinal malignancies (*WHO* and *UEG*) [1]. The five-year survival rate is 90 percent for CRC diagnosed at an early stage compared with 13 percent for those diagnosed later [2]. In this context, immunotherapy could represent an additional strategy to complement surgery, radiotherapy, and chemotherapy to increase the survival of CRC patients, especially when the disease is diagnosed at later stages. Indeed, several immunotherapeutic approaches have been developed for different cancer types, including CRC, with promising clinical results [3]. Immunotherapy for CRC patients includes monoclonal antibodies (mAbs) targeting tumor-associated antigens (TAAs), immune checkpoint inhibitors (ICIs), adoptive cell therapy, anti-cancer vaccines, and oncolytic viruses treatment [3,4,5,6]. However, given the high heterogeneity of CRCs, the therapeutic efficacy of these approaches is variable [7]. In particular, ICI therapy is only effective in a small group of CRC patients characterized by microsatellite instability (MSI) and mismatch-repair deficiency (dMMR), which accounts for less than 20% of patients [7]. Conversely to the other most diffuse types of CRC, marked by mismatch-repair proficiency (pMMR) and microsatellite stability (MSS), MSI and dMMR cancers are usually characterized by enriched immune cell infiltration. The presence of tumor-infiltrating lymphocytes (TILs) has been correlated with the containment of metastases [8], a good clinical outcome [9,10] and a positive response to ICI immunotherapies [11,12,13]. In particular, high numbers of CD8 and CD4 T cells with Th1 profile and NK cells have been correlated with better prognosis in CRC patients [14,15,16]. Based on these clinical data correlating NK cell infiltration with better survival in CRC patients, it is conceivable that novel immune-mediated therapies aimed at increasing the number and/or function of NK cells in tumor lesions could be a useful strategy in CRC containment. In this regard, NK cells, which are innate lymphocytes able to carry out a powerful and immediate response against cancerous cells, may importantly contribute to immune-mediated therapies. NK cells are cytotoxic members of the large and plastic family of Innate Lymphoid Cells (ILCs) [17], whose role in the development of CRC is controversial [18]. NK cells circulate among blood, lymphoid, and non-lymphoid tissues patrolling almost the entire human body, while ILCs are mainly located at mucosal surfaces and are usually not cytolytic [17]. However, NK cells can also be resident in tissues displaying highly variable expression patterns. Thus a clear distinction between tissue-resident NK cells and ILCs, especially ILC1, can be challenging [19,20].

Thanks to their multitude of effector capabilities, including killing activity and modulation of the adaptive immune response, NK cells are ever more considered strategic effectors of the immunotherapeutic approach. In the present review, we discuss the different immunotherapeutic strategies focused on enhancing NK cell function in the fight against CRC.

## 2. Human NK Cells

NK cell capabilities to recognize transformed cells without prior antigen exposure, kill target cells, and release immune-regulating cytokines/chemokines are mainly due to the large repertoire of germline-encoded activating receptors that provide the “on” signal following the interaction with putative ligands (Figure 1). Among all, the non-HLA-specific natural cytotoxicity receptors NKp46, NKp30, and NKp44 (collectively named NCRs), NKG2D, and DNAM-1 are the main activating NK receptors [21,22,23,24,25,26,27].

NCRs belong to the Ig superfamily and possess, in the transmembrane domain, positively charged amino acids that allow association with ITAM-bearing molecules, such as CD3-ζ and/or FcεRI-γ for NKp30 and NKp46 and KARAP/DAP12 for NKp44, and, therefore, the transduction of an activation signal upon ligand recognition [28]. Different membrane-bound, intracellular, and soluble extracellular molecules have been identified as NCRs ligands, including some molecules of viral origin [26,29,30,31,32,33,34,35,36]. 

NKG2D is a type II transmembrane protein with a C-type lectin-like extracellular domain and transduces an activation signal through association with the adapter ITAM-bearing molecule DAP-10 upon recognition of stress-inducible molecules, namely ULBPs and MICA/B. NKG2D ligands can also be shed from tumor-transformed cells and contribute to tumor escape mechanisms [25,37,38].

DNAM-1 is a transmembrane glycoprotein characterized by two extracellular Ig-like domains and by a cytoplasmic portion containing tyrosine residues involved in lymphocyte adhesion and signaling [39]. DNAM-1 recognizes PVR and Nectin-2 [24,40] that are highly expressed on antigen-presenting cells (APCs), tumors, and virus-infected cells. In addition, DNAM-1 shares the same ligands of the TIGIT and CD96 inhibitory receptors but exhibits opposite functions with respect to them, suggesting a complementary role in the regulation of tumor immunity and inflammatory response [40,41].

Another relevant activating receptor is CD16 (FcγRIIIa), a low-affinity receptor for the immunoglobulin G (IgG) Fc fragment, whose binding to opsonized target cells triggers efficient NK cell-mediated killing through antibody-dependent cell-mediated cytotoxicity (ADCC) [42,43]. 

In healthy conditions, all these activating receptors are under the control of inhibitory signals transduced by HLA class I-specific receptors (such as inhibitory KIRs, CD94/NKG2A, LILRB-1) upon recognition of self-HLA molecules [44,45,46] (Figure 1). Mature and functional NK cells usually express at least one inhibitory receptor, specific for self-HLA class I molecules. Indeed, during the cell differentiation process, only NK cells expressing inhibitory receptors recognizing self-HLA class I molecules undergo a process called “education”, consisting of the acquisition of functional competencies in terms of cytotoxicity ability and cytokine secretion [47,48]. This process ensures, on the one hand, self-tolerance towards healthy cells and on the other hand, an efficient response against transformed cells, which usually lack or down-regulate HLA class I expression and acquire or up-regulate the expression of ligands for the non-HLA specific activating NK receptors [49]. NK cells can also express other inhibitory receptors, the non-HLA class I specific receptors, which can regulate the NK cell function by acting as immune checkpoints (ICs), similarly to KIR, NKG2A, and LILRB1. These additional ICs include PD-1, TIM-3, TIGIT, CD96, LAG3, CD161, Siglec-7, and IRp60 and can be up-regulated or de novo expressed by NK cells in pathologic conditions [50,51,52], thus contributing to avoid exacerbated immune responses and also favoring tumor escape (Figure 1).

In this regard, the immunosuppressive CRC tumor microenvironment (TME) due to the activity of tumor cells and the related presence of other immune cells with immune-modulatory properties, such as myeloid-derived suppressor cells (MDSCs) and Th17 cells, can create a disadvantageous milieu affecting the killing properties of NK cells [53]. Several studies have demonstrated that NK cells have reduced functionality in CRC patients [10]. Decreased expression of NKp46 and NKp30 has been demonstrated in CRC-infiltrating NK cells, and the low expression of NKp46 in peripheral blood (PB) NK cells of CRC patients has been correlated with lower relapse-free survival (RFS). Moreover, NKG2D and DNAM-1 expression has been demonstrated to be down-modulated in both PB-NK cells and tissue-infiltrating NK cells of CRC patients [54,55]. 

## 3. Monoclonal Antibodies-Based Treatments to Enhance NK Cell Cytotoxicity

The potential of immunotherapies based on the use of mAbs that target distinct cancer-specific cell markers and may trigger T cell anti-tumor function and NK cell-mediated ADCC has been initially revealed in the hematological setting. Indeed, the successful use of Rituximab, a chimeric anti-CD20 mAb targeting CD20^+^ lymphoma cells [56,57], has prompted the design of different immune tools to also be applied against solid tumors, including CRC. In this context, several mAbs targeting CRC tumor antigens (e.g., EGFR, CEA, Her2) or the TME (e.g., VEGF) that may simultaneously trigger NK cell cytotoxicity have been developed. More recently, the use of mAbs targeting different ICs, represented by inhibitory receptors (e.g., PD-1) that control the function of effector cells, such as T and NK cells, by disrupting inhibitory interactions and restoring anti-tumor capabilities by cytotoxic lymphocytes, has offered unexpected possibilities to cure solid tumors [58,59,60].

### 3.1. Anti-Tumor Associated Antigen (TAA) mAbs, BiKe, and Engagers Enhancing CRC Killing via ADCC

The use of mAbs directed against surface antigens expressed by tumor cells has shown clinical efficacy in different tumors. Treatment with anti-TAA mAbs can induce tumor cell death by several mechanisms, such as directly inducing tumor apoptosis or via ADCC, that is, mediated by CD16 engagement with IgG-opsonized tumor cells. Indeed, the efficacy of an anti-TAA mAb largely employed in the treatment of metastatic CRC, i.e., anti-EGFR (known by the commercial names Cetuximab or Panitumumab or Necitumumab [61]) also relies on enhanced NK-mediated killing via ADCC (Figure 2a). The epidermal growth factor receptor (EGFR; ErbB-1; HER1 in humans), a transmembrane tyrosine kinase receptor, affects cell adhesion, survival and proliferation and is overexpressed in most CRC (75%) and is associated with poor prognosis. Anti-EGFR mAbs can act by blocking ligand binding and thus prevent proliferation in response to EGF. However, in a fraction of CRC patients (36–55% [62]), mutations in KRAS, a signaling molecule downstream of EGFR, render the receptor constitutively active and the anti-EGFR treatment ineffective [63,64,65]. On the other hand, in KRAS-mutated patients, anti-EGFR mAbs could still efficaciously induce ADCC by NK cells as demonstrated in vitro [66] and also in vivo in a recent clinical study (NCT01450319) [67]. Interestingly, this work also reported that anti-EGFR-treated, KRAS-mutant patients carrying homozygous KIR genotypes (AA or BB) have a worse outcome than KIR heterozygotes (AB). The mechanisms underlying this observation are unknown but could be related to altered HLA class-I expression on tumor cells in response to factors, such as IFN-γ, possibly released by CD16-engaged NK cells. 

However, anti-EGFR clinical benefits are confined to a fraction of patients [68], and many efforts to optimize this treatment are in progress. One possibility that has been explored is the manipulation of the Fc fragment to augment its affinity for CD16. For example, a glycoengineered anti-EGFR mAb (GA201, Imgatuzumab, RO5083945) has been demonstrated to induce superior ADCC than the non-manipulated mAb (Cetuximab), allowing both the NK cell impairment often observed in CRC patients [69] and the lower Fc affinity of given CD16 variants that display an amino-acid substitution at position 158 (V158F polymorphism) to be overcome [64,70]. GA201 is not currently employed in clinical trials or therapies, but a similar glycoengineered antibody derived from Cetuximab, Tomuzotuximab, has been tested in combination with a glycoengineered humanized anti-Mucin-1 (MUC-1) antibody (Gatipotuzumab) that targets a tumor-specific epitope of MUC-1, showing promising results in CRC patients [71]. Indeed, Fc-optimized mAbs can elicit potent NK-mediated ADCC and possibly replace the standard antibody in several tumor settings. In this context, anti-EGFR efficacy can be limited not only by KRAS mutations or CD16 variants but also by low Ab penetrance due to TME factors. To overcome these hurdles, novel immune tools such as BiKe and engagers have been designed and have proved their efficacy in vitro and in preclinical models [72,73]. Similar to BiTe (i.e., bispecific single-chain T-cell engager, such as Blinatumomab that couples anti-CD3 to anti-CD20), BiKe are composed of a single-chain variable fragment (scFv) of an antibody specific for a given TAA connected through a short peptide linker to an anti-CD16 scFv, which triggers stronger cytolytic signals in NK cells, favoring the formation of the NK-tumor immunological synapse [35]. Interestingly, a bispecific single domain antibody that targets CD16 on NK cells and recognizes EGFR on tumor cells induces a strong NK cell effector response against both EGFR-expressing CRC cell lines and ex vivo CRC-derived tumor cells, regardless of tumor KRAS mutation status [73]. Moreover, this bispecific antibody induces a consistent release of CXCL10 when ex vivo metastatic CRC tumor cells are co-cultured with autologous PBMC, possibly favoring the in vivo recruitment of effector T and NK cells at the tumor site. Similarly, a tetravalent bispecific (i.e., displaying two binding domains instead of a single chain) fusion antibody targeting CD16 and EGFR on tumor cells also results in highly effective toward a variety of EGFR-expressing tumor cells, regardless of the EGFR expression level [74], thus suggesting its potential efficacy despite CRC heterogeneity.

MAbs and BiKes engaging CD16 to induce ADCC can fail to efficiently trigger NK-mediated cytotoxicity when CD16 is shed from the NK cell surface by matrix metalloproteases such as ADAM17 [75] that can be up-regulated in the TME [76]. Indeed, low CD16 expression levels have been described in tumor-infiltrating NK cells from CRC biopsies [54]. Besides the use of metalloprotease inhibitors to prevent CD16 shedding [77], other immune tools that hold promise to circumvent this issue are trifunctional NK cell engagers (NKCEs). This innovative tool simultaneously targeting NKp46 (or NKp30) and CD16 on NK cells and EGFR on cancer cells has proved to induce superior anti-tumor activity than the standard therapeutic mAbs (e.g., Cetuximab) in preclinical models [72]. Indeed, NKCEs represent plastic tools that could be easily modified with different anti-tumor antigens or anti-NK receptor moieties to better fit both cancer features and NK cell heterogeneity in distinct patients.

Of note, bispecific antibodies simultaneously binding two different tumor antigens that could potently induce NK-mediated ADCC are currently in clinical trials for advanced CRC or in preclinical models, such as anti-EGFR and anti-c-MET (Mesenchymal Epithelial Transition receptor, highly expressed or amplified in subsets of CRC, NCT04930432) or anti-EGFR and anti-LGR5 (cancer stem cell marker) [78].

Beyond EGFR, several other TAAs associated with CRC, such as MUC-1, CEA, gpa33, HER2, PD-L1, and CD73, are currently being explored as targets of mAb-mediated immunotherapy, possibly benefitting NK cell ADCC [53,79]

### 3.2. Immune Checkpoint Inhibitors (ICI) to Unleash NK Cell Killing against CRC

In the TME, NK cell function can be dampened mainly by the down-modulation of activating receptors in response to soluble factors released by tumor or tumor-associated cells (e.g., TGF-β, PGE2, soluble B7-H6, IDO1-derived catabolites, soluble ligands [10,80]) and/or by the engagement of ICs expressed by NK cells. Some studies have shown that NK cells infiltrating CRC can express multiple ICs, including both inhibitory receptors specific for HLA class I molecules (NKG2A and KIRs) and those recognizing non-HLA class I ligands (e.g., PD-1, TIM-3, LAG3, TIGIT) [54,81,82,83]. Further studies investigating IC expression and function in CRC-infiltrating NK cells are needed; however, the use of different ICIs that block the interactions between ICs and their ligands expressed on tumor cells is a promising immunotherapeutic approach that could be capable not only of restoring T cell immunity, but also of unleashing NK cell anti-tumor potential (Figure 2b). 

A large number of clinical trials in metastatic or advanced MSI and dMMR CRC have been based on the administration of anti-PD-1 and/or anti-PD-L1 mAbs. Interestingly, recent data reported that MSS patients with proficient MMR but harboring the POLE mutation (pole encodes the DNA polymerase responsible for lead strand DNA replication) that favors high neoantigens generation, high TMB, and recruitment of TILs, including NK cells, can also benefit from anti-IC therapies [7,84]. Indeed, with the aim of optimizing/enlarging the ICI potential for MSS patients or ICI refractory/resistant MSI patients, several currently active (or still recruiting) clinical trials are exploring the effect of anti-PD-1 mAbs in combination with chemotherapy or with kinase inhibitors or with other mAbs, such as anti-VEGF, both in MSI and in MSS CRC at advanced stages. Other trials are aimed at simultaneously blocking multiple ICs in combination with conventional chemotherapies that could contribute to converting “cold” tumors to immune-active tumors sensitive to ICI therapies [85]. Although the PD-1-PD-L1 axis may actually contribute to hamper NK cell function in CRC, a major role is likely played by the IC NKG2A, which is consistently expressed by both NK and T cells in CRC TILs [54,81,86]. NKG2A recognizes the non-classical HLA class I molecule HLA-E, which results as overexpressed in a fraction of CRC, preferentially in MSI compared to MSS, and whose expression is associated with a worse prognosis [81,87,88]. Indeed, a combination of anti-NKG2A (Monalizumab or IPH2202) and anti-PD-L1 (Durvalumab) mAbs, offered to patients with metastatic MSS CRC, showed promising activity in a recent clinical trial (NCT02671435) [89]. An additional promising strategy is based on combining the blockade of inhibitory signals with the delivery of activating signals. In this context, in vitro data demonstrated that NKG2A blockade with Monalizumab boosts NK cell-mediated ADCC against Cetuximab-coated tumor targets [81]. Indeed, this combination was effective in a phase 1–2 trial and a phase 3 trial (NCT04590963) is ongoing in a Squamous Cell Carcinoma of Head and Neck (SCCHN) cohort. Along this line, a phase 1 trial (NCT05162755) that combines an NKG2A and PD-1 blockade with anti-EGFR is currently recruiting patients with metastatic gastric tumor and CRC. A similar strategy can be pursued by novel immune tools designed to stimulate ADCC via CD16 and simultaneously block PD-1/PD-L1 interactions. These molecules have been engineered to incorporate an IL-15 moiety with the aim of promoting NK cell activation, in vivo persistence and proliferation and have shown promising results in vitro and in preclinical models [90,91]. 

Finally, the effects of other ICI on NK cell function, such as those blocking LAG3 and TIGIT that can be expressed by NK cells in CRC, deserve to be examined [59,79].

## 4. Cytokines Enhancing NK Cells Mediated Killing of CRC

Therapeutic approaches based on the use of cytokines, mainly IL-2 and IL-15, able to directly stimulate and promote NK cell activation, persistence, and expansion, have been tested in several preclinical and clinical studies [36,92,93]. In vitro results demonstrated that the combined use of IL-2 and IL-15 with Cetuximab can improve the impaired ADCC response of NK cells derived from CRC patients with respect to Cetuximab alone [94]. In addition, in CRC patients, IL-15 was shown to recover anti-tumor functions of infiltrating NK cells in liver metastases [95]. Actually, Aldesleukin (IL-2) is used in clinical trials testing vaccines in patients with metastatic CRC (NCT00019591). 

However, the use of both cytokines in a clinical setting is related to severe adverse events (AEs), affecting the safety and efficacy of the treatment [96,97,98]. Re-engineering these two cytokines in order to reduce toxicity effects and preserve biological properties is an actual challenge. 

Interestingly, in a recent paper, Silva and colleagues developed a computational protein design method and generated a neo-protein mimicking IL-2 with preserved functional properties and the ability to bind the heterodimeric IL2Rβγ receptor but not IL-2Rα [99] (Figure 2c). This property can overstep one of the issues concerning IL-2 administration, which is the production of immunosuppressive TGF-β by IL-2/IL-2Rα-mediated activation of regulatory T cells (Treg) that constitutively express IL-2Rα at high levels [100]. The authors showed that the newly engineered IL-2 caused reduced expansion of immunosuppressive T cells (Treg) in comparison to native IL-2 and led to a dose-dependent delay in tumor growth, both in colon cancer and melanoma mouse models, without immunogenicity and with reduced toxicity [99]. The other two modified IL-2 products, IL-2 superkine and NKTR-214, derived by sequence mutation and PEGylation of native IL-2, respectively [101,102], have been generated with the aim of reducing IL2/IL-2Rα interaction. IL-2 superkine has been demonstrated to decrease Treg immunosuppressive activity and improve anti-tumor responses in vivo against a murine colon carcinoma model. Similarly, in melanoma mouse models, treatment with NKTR-214 resulted in Treg number reduction in comparison to IL-2 without affecting CD8 and NK responses. In addition, the authors demonstrated high efficacy of NKTR-214 administration combined with anti-CTLA-4 in reducing murine colon tumor growth in syngeneic models without evident adverse signs. NKTR-214 revealed good tolerability in a completed clinical trial that had enrolled patients with advanced renal carcinoma and melanoma [103] and is under evaluation in other clinical trials as monotherapy or combined treatment.

Regarding the modified IL-15-derived products, N-803 (formerly known as ALT-803), a compound developed by binding an IL-15 mutant to a soluble dimeric IL-15Ra-Fc fusion protein, has been proven capable of expanding NK and CD8 T cells and of inducing a significantly higher anti-tumor activity than rIL-15. In addition, N-803 treatment could also prolong survival against pulmonary metastasis, and it has shown a synergistic effect in combination with ICI, mainly with anti-CTLA-4, in tumor models generated using colon carcinoma cells [104,105]. In the hematological setting, N-803 initially provided good results in patients who relapsed after transplantation [106]. However, more recent studies highlighted that N-803 could promote allogeneic cell rejection by host T cells, limiting the in vivo persistence of adoptive NK cells and overall clinical responses to allogeneic adoptive cell therapy [107]. Thus, whether N-803 could be the optimal cytokine support to promote NK cell expansion and persistence after the adoptive transfer has still to be defined [108], and its use deserves further studies in solid tumor treatment.

## 5. Adoptive Transfer of NK Cells 

### 5.1. Unmodified NK Cells for the Treatment of Solid Tumors

Already, in the 90s, first evidence demonstrated the feasibility of administering expanded autologous NK cells in combination with cytokines (e.g., IL-2) in the treatment of cancer patients (e.g., breast cancer and lymphoma patients) [109,110] and later on different studies demonstrated the beneficial effect of therapies using allogeneic NK cell infusion in transplantation and non-transplantation settings [111,112,113,114,115,116]. Most studies and clinical trials using NK cell-based product administration (allogeneic NK cells and NK-92 cell lines) were focused on the treatment of hematological malignancies. Less explored was the effectiveness of adoptive NK cell treatments for solid tumors. 

The most recent strategies using adoptive NK cell administration take advantage of recent advances in expansion, activation, and cryopreservation of NK cells, as well as the enrichment of NK cell sources. Indeed, the NK-92 cell line, PB cells, umbilical cord blood (UCB), and induced pluripotent stem cells (iPSCs) have been demonstrated to be supplies for NK adoptive transfer, but each of them is characterized by lights and shadows [117,118]. The unmodified but irradiated NK-92 cell line, approved by the FDA for adoptive transfer in clinical trials [119], has some limitations, mainly due to the reduced persistence in vivo after the infusion, the lack of CD16 impairing triggering via ADCC, and the lack of NKp44 expression [120,121], compromising the natural cytotoxicity response. UCB-derived NK cells can be rapidly available, and show reduced GvHD and viral transmission risk, but few cells can be obtained from UCB [122]. PB-derived NK cells can be easily provided, and require activation and expansion, but the yield could be affected by donor variability. IPSCs can be maintained in an undifferentiated state and grown indefinitely, therefore enabling the production of huge numbers of homogeneous NK cells providing a base for a standardized, off-the-shelf approach [123,124,125].

From the beginning of 2019, the clinical trial NCT03841110 has been assessing, for the first time, the safety of iPSC-derived NK cells in the treatment of lymphoma and advanced solid tumors, including CRC. The treatment consists of lympho-conditioning with fludarabine and cyclophosphamide followed by the administration of FT500, an iPSC-derived NK cell product, as monotherapy or combined with 1 of 3 approved ICIs (Nivolumab, Pembrolizumab, or Atezolizumab), in patients who have failed prior ICI therapy [126]. Published results showed that 69% of solid tumor patients (n = 13) had the best response of stable disease (evaluated by iRECIST), no AEs related to cytokine therapy, and no immune-mediated toxicity as GvHD, cytokine release syndrome (CRS), or neurotoxicity (NT). A subsequent observational study (NCT04106167) is designed to provide long-term safety, efficacy, and survival data for subjects who took part in the interventional study of allogeneic FT500 NK cellular immunotherapy. 

Moreover, another iPSC-derived NK cell product is under evaluation in a recently opened clinical trial (NCT05069935), which enrolls patients with advanced solid tumors, including metastatic CRC. The study is finalized to evaluate the right dose of the allogeneic NK cell FT538 product. In this case, the iPSC-derived NK cells platform is engineered to be CD38 knock-out and to express a high affinity, non-cleavable version of the Fc receptor CD16a and a membrane-bound IL-15/IL-15R fusion protein. These modifications are focused on enhancing ADCC capability and in vivo persistence [127,128,129,130]. 

Another active clinical trial (NCT03319459) enrolling patients affected by advanced solid tumors expressing EGFR, including CRC patients, is evaluating the effect of PB-derived allogeneic NK cell administration in combination with Cetuximab. These infused allogeneic NK cells have been enriched in highly functional mature NK cells. Indeed, this trial is based on results obtained by Cichocki and colleagues, who demonstrated that exposure to IL-15 and to an inhibitor of Glycogen synthase kinase (GSK) 3 enhanced NK cell-mediated anti-tumor activity and ADCC against tumors of different histotypes when administered in combination with a mAb against EGFR or other TAAs [131].

### 5.2. Engineered NK Cells as Therapy for Colorectal Cancer Patients

Alongside the T cell-modifying strategies, innate immune cell engineering has been more recently considered a promising therapeutic approach to counteract CRC. Chimeric Antigen Receptors (CARs) are engineered proteins composed of an extracellular antigen-binding domain targeting TAA, a transmembrane region and an intracellular activating signaling domain. A large plethora of constructs (from first to fourth generation) has been developed in order to optimize the receptor function, mainly modifying the intracellular sequence with different stimulatory or co-stimulatory domains, and to strengthen T and NK cell response [132,133]. Nowadays, different clinical and preclinical studies report encouraging results on the use of CAR-T, especially against hematological malignancies [134,135,136]. However, the use of CAR-NK cells could offer several advantages over their CAR-T counterpart [137]. Indeed, CAR-NK cells (1) can be prepared in advance and from different sources (NK-92 cell line, PB cells, UCB, and iPSC); (2) have been demonstrated to be able to kill in a CAR-dependent and independent manner, overtaking possible tumor escape mechanisms, such as loss of the antigen recognized by the CAR; and (3) are less capable of inducing CRS and neurotoxicity therapy-related events. 

Inducing overexpression of activating receptors can improve NK anti-tumor activity (Figure 2d and Table 1). To this end, in 2019, Xiao and coworkers demonstrated that intra-peritoneal injection of short-lived PB-derived CAR-NK cells, generated by RNA electroporation with a construct coding for NKG2D extracellular domain combined to DAP12 signaling moiety (NKG2D CAR-NK), significantly reduced tumor burden and progression in xenograft mice generated with human CRC cell lines [138]. In addition, in the related pilot clinical trial (NCT03415100), three patients with refractory metastatic CRC were treated with a local infusion of NKG2D CAR-NK (one in autologous and two in haploidentical setting) with benefit and without severe AEs. In particular, in two patients, the number of cancer cells in ascites fluid decreased, and, in another one, a complete metabolic response in liver metastasis was observed. These encouraging results show that localized solid tumor treatment with CAR-NK cells may be a therapeutic strategy to pursue. However, further analyses are needed to assess the persistence of the treatment outcome. In this regard, additional informative results will also be obtained from another clinical trial (NCT05213195), which has recently started and that evaluates the effects of intra-peritoneal and intra-venous NKG2D-CAR-NK infusion in patients with refractory metastatic CRC. Other interesting results aimed at increasing NKG2D-mediated killing activity of NK cells against CRC liver metastases have been carried out from preclinical studies focused on the analysis of the anti-tumor activity of allogeneic healthy donors’ NK cells modified with a chimeric NKG2D receptor fused to co-stimulatory (OX40) and signaling (CD3ζ) domains (to enhance their intrinsic activity) and equipped with membrane-bound IL-15 (to enhance in vivo persistence) [139,140]. Indeed, NKG2D-CAR-NK cells are shown to perform enhanced in vivo cytotoxicity against hepatocellular carcinoma cells in a SNU449 (HCC cell line) xenograft model [141]. 

Preclinical/clinical studies are developing an immune-mediated therapeutic intervention to also target EpCAM positive cells since high expression of EpCAM (CD326) is one of the most common alterations in solid tumors of epithelial origin, including CRC (NCT03013712) [147,148]. A recent preclinical study analyzed the effect of NK-92 modified with a second-generation CAR, targeting EpCAM, in the control/eradication of the CRC line HCT-8-Luc in a subcutaneous xenograft NOD/SCID mice model and demonstrated that CAR-NK-92 cells significantly reduced tumor growth compared to the control NK-92 cell line [142]. The anti-tumor response could also be incremented by the combined use of Regorafenib, a multikinase inhibitor with activity against different protein kinases involved in oncogenesis [149,150] and successfully used to treat refractory metastatic CRC [151]. 

CEA is considered another molecule to direct targeted therapy (NCT02349724) [152] since it is widely expressed in CRC tissues and scarcely expressed in normal adult tissues (lung cells and GI-epithelial cells). In a preclinical study, anti-CEA CAR NK-92MI, an IL-2 independent derived NK-92 cell line, has been demonstrated to recognize and kill CEA-expressing tumor cells at high and moderate levels [143]. Since chemotherapy frequently induces up-regulation of CEA, anti-CEA modified NK cells could be a secondary rescue line of intervention in the treatment of refractory CRC. However, further evaluation on CEA cell-mediated targeting will be needed since clinical results obtained by targeting CEA with CAR-T cells have shown high toxicity, may be related not only to the CAR-T-induced cytokine storm upon recognition of antigen on tumor but also to non-tumor tissues (NCT01212887) [153]. 

Actually, a broader set of molecules is being studied as CAR targets for CRC treatment, and, among them, MUC-1 and HER-2 appear as suitable targets for cell-mediated therapy (Table 1).

Further, MUC-1 is highly expressed in CRC cells, and increasing evidence suggests this highly glycosylated protein is a potent target for diverse immunotherapy strategies, including the generation of modified NK cells equipped with anti-MUC-1 CAR. In this line, a phase I/II clinical trial (NCT02839954) is evaluating the safety and effectiveness of anti-MUC-1 CAR NK cell immunotherapy in patients with MUC-1^+^ relapsed or refractory solid tumors [144]. 

HER-2 belongs to the EGFR family, and its overexpression is correlated with the stage of disease and reduced survival in CRC [154,155] as well as in breast cancer and gastric adenocarcinomas [156,157,158]. Further, an NK-92-derived product is under investigation for a clinical translation aimed at treating HER2-expressing solid tumors (NCT04319757). Indeed, it has been recently shown that an adapted subpopulation of NK-92 expressing functional endogenous CD16 and further modified with the conjugation with Trastuzumab, an anti-human epidermal growth factor receptor 2 (HER-2) antibody, displayed enhanced cytotoxicity against HER-2-positive targets in vivo and in vitro. The absence of cell manipulation with viral vector or transposon systems, possibly mediating viral insertion mutation and imprecise chromosomal insertion, respectively, could represent an advantage with respect to other engineered products [145,146].

## 6. Future Perspectives

The combination of different strategies to fully unleash cytotoxic immune cell function is certainly one of the most promising approaches to increase their impact in anti-tumor immunotherapies. For example, the combination of immunotherapies aimed at reducing the suppressive effect exerted by TME-related factors with ICI or triggering mAbs holds promise to increase NK and T cells’ anti-tumor potential. In this context, the adenosine signaling pathway has quite recently emerged to be a target for immunotherapy in the treatment of solid tumors. Indeed, adenosine represents an immune-suppressive modulator impairing CD8 T and NK cell anti-cancer immune response. In particular, the ecto-enzyme CD73, which is overexpressed in response to hypoxia signaling and, together with CD39, converts extracellular ATP to adenosine, represents a very promising candidate [159,160,161,162,163,164,165,166,167]. A recent paper by Kim et al. suggested that blocking ATP/adenosine signaling in combination with PD-1 inhibitors results in a synergistic approach with the potential to improve the treatment of refractory CRC [168] and refractory renal cell cancer [169]. In this regard, in the preclinical setting, CD73 blockade has been demonstrated to enhance the ability of NKG2D CAR-engineered NK-92 cells to restrain tumor growth in lung cancer xenograft models [170]. Besides inhibiting adenosine production, blocking the activity of IDO1, an enzyme overexpressed in CRC and other tumors, which converts tryptophan (Trp) in immunosuppressive catabolites, could be a promising strategy [171]. In particular, Trp-derived kynurenine limits T cell proliferation and NK cell cytotoxicity by down-modulating the activating receptors NKp46 and NKG2D [172]. Along this line, two trials (NCT02178722, NCT02959437) have explored the combined effect of PD-1 blockade and IDO1 inhibition in MSS patients and could provide interesting hints to optimize immune treatments in solid tumors by rescuing NK and T cell effector function.

Exploring new possibilities to potentiate NK cell function in vitro before infusion in adoptive cell therapy is another open field of investigation. Romee and colleagues pre-activating ex vivo allogeneic NK cells with a mixture of cytokines, including IL-12, IL-15, and IL-18, characterized the so-called “cytokine induced memory-like NK cells” (CIML-NK), which show enhanced proliferation capability, high IFN-γ production, high cytotoxicity, and, after a second encounter with tumor targets, an enhanced recall response [173,174]. CIML-NK cells appear as a promising NK cell therapy and clinical trials evaluating the use of CIML-NK for refractory/relapsed AML patients showed encouraging results, prompting the design of clinical trials evaluating the use of CIML-NK in solid tumors [175]. Moreover, the CIML-NK platform could be further combined with other immune-mediated strategies, such as CAR engineering. In this regard, recent studies on CAR-engineered CIML for targeting resistant B lymphoma and AML demonstrate the feasibility of this alternative approach for cancer immunotherapy [176,177]. Interestingly, a novel triple-cytokine fusion molecule has been recently created (named 18/12/TxMin), composed of the scaffold of N-803 linked with IL-18 and the IL-12 p70 single-chain. This novel molecule exactly mimics the effect of exposing NK cells to the combination of individual cytokines both in vitro and in vivo and could represent a more suitable tool to activate and expand CIML-NK cells for adoptive therapy purposes and combinatorial therapies [178]. In addition to cytokine exposure or mAb-mediated triggering, NK cell function can be greatly potentiated through Toll-like receptor engagement [179,180,181]. Along this line, a recent study by Long and colleagues demonstrated that NK cells exposed to Cetuximab in combination with oncolytic reovirus were activated mainly by TLR-3 and showed increased ADCC against CRC lines in vitro independently to KRAS mutation status and EGFR expression level. Importantly, the combined exposure to reovirus and Cetuximab provided a greater NK-mediated anti-tumor effect than monotherapy with Cetuximab in vivo [182].

Another real challenge in NK-mediated immunotherapies for solid tumors is to increase not only NK cell effectiveness but also their abundance in the TME [183]. Up-regulating chemokine receptors on NK cells can be a valuable option to achieve this goal in CRC. In a recent study, the up-regulation of the chemokine receptor CXC chemokine receptor 2 (CXCR2) and IL-2 expression on NK-92 cells by CRISPR-Cas9 gene-editing has been shown to increase migration into tumor sites and induce stronger cell-killing and proliferation activity of engineered cells. The benefit of increased recruitment of gene-edited NK-92 cells was also confirmed by better control of tumor growth in vivo [184].

Finally, identifying novel markers and up-regulated or dis-regulated pathways during neoplastic transformation is paramount to designing new specific mAbs to directly target cancer cells but also to develop combined therapies taking advantage of NK-mediated ADCC response. 

Several immunotherapeutic strategies against CRC involving NK cell activity are, therefore, under evaluation and are opening interesting perspectives. However, a deeper understanding of the immune landscape in different types of primary CRC and metastatic lesions, including a sharp analysis of NK cell signature and function, will further contribute to better design of novel immunotherapies. 

## Figures and Tables

**Figure 1 vaccines-10-01033-f001:**
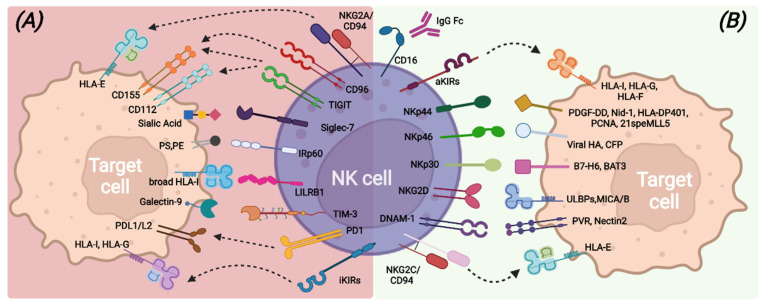
Main inhibitory (**A**) and activating (**B**) NK-cell surface receptors and their cognate ligands on target cells.

**Figure 2 vaccines-10-01033-f002:**
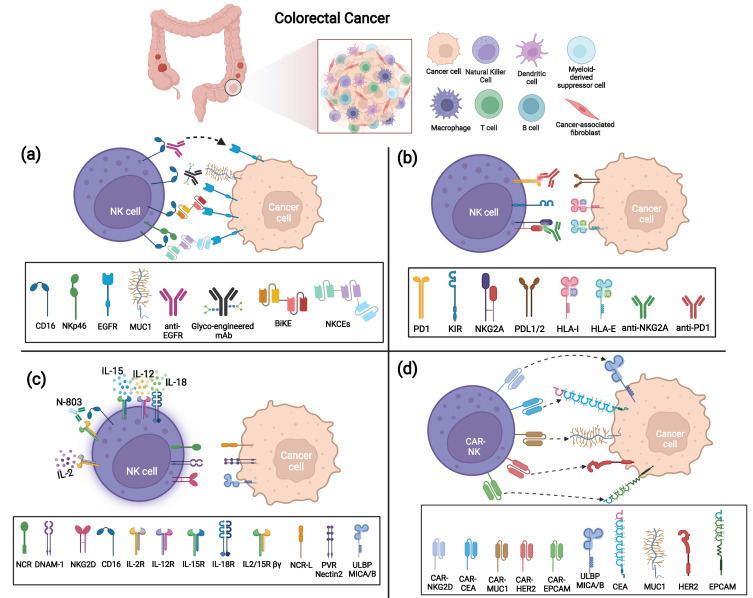
Strategies to enhance anti-tumor NK cell function against Colorectal Cancer (CRC). (**a**) ADCC triggering strategies via anti-TAA mAbs, BiKe, or NKCEs; (**b**) IC blockade via anti-NKG2A and anti-PD-1 mAbs; (**c**) Cytokine activation of NK cells by IL-2, IL-15/IL-12/IL-18 combination, or N-803; (**d**) CAR-modified NK cells targeting several TAAs (CEA, MUC-1, EpCAM, HER-2, and NKG2D ligands MICA/B and ULBPs).

**Table 1 vaccines-10-01033-t001:** Current preclinical studies describing engineered NK cells to target CRC.

Targeted Molecules	NK Cell Sources	Engineered Constructs	References
NKG2D-Ls	Peripheral Blood	NKG2D-CD8-DAP12	[138]
NKG2D-Ls	Peripheral Blood	NKG2D-OX40-CD3ζ and mbIL-15	[139,140]
Ep-CAM	NK-92	anti-EpCAMscFv-CD8-4-1BB-CD3ζ	[142]
CEA	NK-92	anti-CEAscFV-CD8α-CD3ζ	[143]
MUC-1	NK-92	anti-MUC1-pNK	[144]
HER-2	NK-92	ACE-oNK-HER-2	[145,146]

## Data Availability

Not applicable.

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
