# Peer review of "NK Cell-Based Immunotherapy in Colorectal Cancer"

_vaccines, 2022, doi:10.3390/vaccines10071033_

Round 1

Reviewer 1 Report

In this manuscript, Chiesa et al. give an overall review on the role of NK cells in colorectal cancer and the potential application for these cells in anticancer therapies. The manuscript is comprehensive, well written and interesting especially for the researchers within the tumor immunobiology and immunotherapy fields. The only suggestion I would make is providing a table summarizing the information presented in Section 5.2. Also, the text needs some minor editing for the English grammar and punctuation.

Author Response

In this manuscript, Chiesa et al. give an overall review on the role of NK cells in colorectal cancer and the potential application for these cells in anticancer therapies. The manuscript is comprehensive, well written and interesting especially for the researchers within the tumor immunobiology and immunotherapy fields. The only suggestion I would make is providing a table summarizing the information presented in Section 5.2. Also, the text needs some minor editing for the English grammar and punctuation.

We thank the Reviewer for the positive evaluation of our work.

According to the Reviewer’s suggestion we have added a table summarizing the information presented in Section 5.2 and edited the text to the best of our abilities.

Reviewer 2 Report

The manuscript describes the role and functions of anti-tumor effect of NK cells in TME. This is well-written as a short format review article. In the manuscript, characteristic features of anti-cancer NK cells, supporting T cell responses, utilizing antibody dependent cell killing in cancer immunotherapy are well summarized. I would like to add some comments, which, in my opinion need to be addressed to improve the manuscript. 

1.     An additional figure to the section 2, in addition to the description in the main text, displaying receptors and ligands for activating and inhibiting NK cells for their action for killing target cells would be helpful for better understanding.

2.     Please rationalize more detailed necessity of NK cell-mediated immunotherapy specifically in CRC. Main text does not limit reporting studies to CRC, but also summarize other cancer types. It is to recommend to categorize cancer types based on the response to NK cell-mediated immunotherapy or to modify the title.

3.     The first part of the section 6 (Future perspective), explaining immune-suppressive adenosine signaling pathway would be better fitting to introduction section. 

Author Response

The manuscript describes the role and functions of anti-tumor effect of NK cells in TME. This is well-written as a short format review article. In the manuscript, characteristic features of anti-cancer NK cells, supporting T cell responses, utilizing antibody dependent cell killing in cancer immunotherapy are well summarized. I would like to add some comments, which, in my opinion need to be addressed to improve the manuscript. 

  1. An additional figure to the section 2, in addition to the description in the main text, displaying receptors and ligands for activating and inhibiting NK cells for their action for killing target cells would be helpful for better understanding.

According to the Reviewer’s suggestion we have added a new figure (figure 1) in section 2 depicting the main inhibitory and activating NK receptors and their respective ligands.

  1. Please rationalize more detailed necessity of NK cell-mediated immunotherapy specifically in CRC. Main text does not limit reporting studies to CRC, but also summarize other cancer types. It is to recommend to categorize cancer types based on the response to NK cell-mediated immunotherapy or to modify the title.

We thank the Reviewer for His/Her observations.

According to His/Her requests we have modified a sentence to better clarify the need for NK cell-mediated immunotherapy in CRC in the introduction section where this concept had been already mentioned.

We agree with the Reviewer that our manuscript also mentions a few studies related to tumors different from CRC. However, we have discussed data regarding other tumor types with the aim to suggest that a given (NK-based) treatment successfully employed to cure non CRC cancers could provide benefits also in CRC patients. In addition since several immunotherapies are under testing in clinical trials offered to different tumor patients, not only to CRC patients, we believe it’s worth mentioning all cancer types examined and evaluate the results obtained in a comprehensive manner that could capitalize data from all the different tumor settings.

However, our aim is keeping the focus on CRC, we have thus revised the manuscript eliminating the discussion related to other tumor types when not necessary.

  1. The first part of the section 6 (Future perspective), explaining immune-suppressive adenosine signaling pathway would be better fitting to introduction section. 

We thank the Reviewer for His/Her observations. We agree that a mechanicistic explanation would fit better in another section. However we’d rather describe the relevance of therapies targeting the immune-suppressive adenosine signaling pathway combined to other novel immunotherapies such as CAR-therapy or ICI in the future perspectives section since these combinations really represent a novel promising strategy to pursue in the future. We hope the Reviewer might understand our point of view. 

Reviewer 3 Report

Major Comments:

This manuscript deals with NK cell-based immunotherapy in colorectal cancer. It provides an overview on the topic and on current developments, including an illustrative figure. Alas, application of this knowledge in routine clinical management is still limited; however, this is, of course, not the fault of the authors; they report the present state of research. For the time being, many options and future perspectives remain theoretical. In this context, this paper may also be seen as an incentive to further investigations.

Additional Comments/Suggestions:

Lines 238-240: "Therapies (https://www.clinicaltrials.gov/ct2/results?cond=Colorectal+Cancer&term=PD1&cntry=&state=&city=&dist)" – I would propose using a citation number and adding this internet address to the reference list. The same applies to line 409.

Lines 260-261: "Finally, the effect of other ICI on NK cell function, such as those blocking LAG3 and TIGIT that can be expressed by NK cells in CRC deserve to be examined [58,78]." -> Finally, the effects of other ICI on NK cell function, such as those blocking LAG3 and TIGIT that can be expressed by NK cells in CRC deserve to be examined [58,78].

Lines 296-297: "NKTR-214 revealed well tolerability in a completed clinical trial…" -> NKTR-214 revealed good tolerability in a completed clinical trial…

Line 362: "allogeneic NK cells administration" -> allogeneic NK cell administration (or: administration of allogeneic NK cells). This principle also applies to other word groups used in this text.

Line 443: "Also MUC-1 is highly expressed on CRC cells and increasing evidences suggest this highly glycosylated protein as a potent target for diverse immunotherapy strategies..." -> Also MUC-1 is highly expressed on CRC cells and increasing evidence suggests this highly glycosylated protein as a potent target for diverse immunotherapy strategies...

Punctuation:

Line 148: "…treatment [62–64] .On…" -> …treatment [62–64]. On…

Line 204: "…anti-LGR5 (cancer stem cell marker)[77]" -> …anti-LGR5 (cancer stem cell marker) [77].

Line 316: "Already in the 90’s" – I would prefer: Already in the 90s (or: Already in the ´90s).

Note: The first question of the "Review Report Form" ("Is the work a significant contribution to the field?") is not applicable (as this manuscript provides an overview on the field and not a "contribution" per se). It was not possible to skip this question; therefore, three stars (average) were marked, although, if possible, the appropriate answer would be "N/A".

Author Response

This manuscript deals with NK cell-based immunotherapy in colorectal cancer. It provides an overview on the topic and on current developments, including an illustrative figure. Alas, application of this knowledge in routine clinical management is still limited; however, this is, of course, not the fault of the authors; they report the present state of research. For the time being, many options and future perspectives remain theoretical. In this context, this paper may also be seen as an incentive to further investigations.

Additional Comments/Suggestions:

Lines 238-240: "Therapies (https://www.clinicaltrials.gov/ct2/results?cond=Colorectal+Cancer&term=PD1&cntry=&state=&city=&dist)" – I would propose using a citation number and adding this internet address to the reference list (Ref.#85). The same applies to line 409.

Lines 260-261: "Finally, the effect of other ICI on NK cell function, such as those blocking LAG3 and TIGIT that can be expressed by NK cells in CRC deserve to be examined [58,78]." -> Finally, the effects of other ICI on NK cell function, such as those blocking LAG3 and TIGIT that can be expressed by NK cells in CRC deserve to be examined [58,78].

Lines 296-297: "NKTR-214 revealed well tolerability in a completed clinical trial…" -> NKTR-214 revealed good tolerability in a completed clinical trial…

Line 362: "allogeneic NK cells administration" -> allogeneic NK cell administration (or: administration of allogeneic NK cells). This principle also applies to other word groups used in this text.

Line 443: "Also MUC-1 is highly expressed on CRC cells and increasing evidences suggest this highly glycosylated protein as a potent target for diverse immunotherapy strategies..." -> Also MUC-1 is highly expressed on CRC cells and increasing evidence suggests this highly glycosylated protein as a potent target for diverse immunotherapy strategies...

Punctuation:

Line 148: "…treatment [62–64] .On…" -> …treatment [62–64]. On…

Line 204: "…anti-LGR5 (cancer stem cell marker)[77]" -> …anti-LGR5 (cancer stem cell marker) [77].

Line 316: "Already in the 90’s" – I would prefer: Already in the 90s (or: Already in the ´90s).

We thank the Reviewer for the positive evaluation of our work.

According to the Reviewer’s suggestions we have added the internet addresses to the reference list, modified the sentences in the text as indicated and corrected the puntuaction.